# Clinical and Molecular Characteristics and Outcome of Adult Medulloblastoma at a Tertiary Cancer Center

**DOI:** 10.3390/cancers16213609

**Published:** 2024-10-25

**Authors:** Abdelatif Almousa, Ayah Erjan, Nasim Sarhan, Mouness Obeidat, Amer Alshorbaji, Rula Amarin, Tala Alawabdeh, Ramiz Abu-Hijlih, Mohammad Mujlli, Ahmad Kh. Ibrahimi, Dima Abu Laban, Bayan Maraqa, Abdallah Al-Ani, Sarah Al Sharie, Maysa Al-Hussaini

**Affiliations:** 1Department of Radiation Oncology, King Hussein Cancer Center, Amman 11941, Jordan; aalmousa@khcc.jo (A.A.); ae.12479@khcc.jo (A.E.); nsarhan@khcc.jo (N.S.); rhijlih@khcc.jo (R.A.-H.); albrahimi@khcc.jo (A.K.I.); 2Department of Neurosurgery, King Hussein Cancer Center, Amman 11941, Jordan; mo.15790@khcc.jo (M.O.); dr.shurbaji@hotmail.com (A.A.); 3Department of Neuro-Oncology, King Hussein Cancer Center, Amman 11941, Jordan; ramarin@khcc.jo (R.A.); ta.11388@khcc.jo (T.A.); 4Department of Neuro-Radiology, King Hussein Cancer Center, Amman 11941, Jordan; mm.13631@khcc.jo (M.M.); da.11945@khcc.jo (D.A.L.); 5Department of Pathology and Laboratory Medicine, King Hussein Cancer Center, Amman 11941, Jordan; bm.11034@khcc.jo; 6Office of Scientific Affairs and Research, King Hussein Cancer Center, Amman 11941, Jordan; abdallahalany@gmail.com (A.A.-A.); sarahalsharie2000@gmail.com (S.A.S.); 7Department of Cell Therapy and Applied Genomics, King Hussein Cancer Center, Amman 11941, Jordan

**Keywords:** adult, medulloblastoma, radiation, outcome, molecular subgrouping

## Abstract

Adult medulloblastoma is an uncommon brain tumor, distinct from its pediatric counterpart in clinical presentation and molecular characteristics. Its management often relies on treatment strategies derived from pediatric cases due to limited research on adults. Our study evaluates the clinical and molecular characteristics of 53 adult patients treated at a single center and explores factors influencing survival outcomes. We found that the extent of surgery and disease stage significantly impacted survival, while molecular subtypes did not correlate with prognosis. High-risk patients exhibited poor outcomes, suggesting a need for more aggressive treatment approaches. Understanding these factors is crucial for improving survival rates in adult patients.

## 1. Introduction

Medulloblastoma is the most common malignant brain tumor in children [1]. In adults, the incidence of medulloblastoma is rare, accounting for fewer than 1% of primary intracranial malignancies [2]. A recent systematic review and meta-analysis showed that the global prevalence of medulloblastoma is 7.7% [3]. The annual incidence of medulloblastoma in adults is roughly 0.6 per 1,000,000, which is much lower than that observed in children (4.1 per 1,000,000) [4,5]. Adult medulloblastoma is associated with a median age at diagnosis of 30 years and a male-to-female predominance of three to two [6]. It is distinct from its pediatric counterpart in terms of genomic aberrations, histopathologic make-up, and clinical characteristics [7,8], all of which lead to different prognostic outcomes. For instance, Sonic Hedgehog (SHH)-activated medulloblastomas are considered the most common subtype in adults, accounting for more than two-thirds of cases, while other subtypes are less frequent [9,10,11]. In comparison, SHH-activated subgroups account for 64% in the age group 0–3 years, 16% in 3–10 years, and 17% in 10–18 years [10,11].

Due to the low incidence and lack of prospective studies of adult medulloblastomas, treatment standards are not well established but, rather, are extrapolated from pediatric data [9,12]. Current management of adult medulloblastoma includes maximum surgical resection followed by craniospinal irradiation (CSI) with or without chemotherapy, depending on risk stratification [6,9]. The issues associated with this conventional treatment approach include the lack of validation, paucity of level I evidence of the efficacy of chemotherapy, and variance of treatment outcome in patients with similar clinical risk [6]. The latter is mostly attributed to variance in tumor biology.

The current risk-stratification models for adult medulloblastoma are based on clinical variables [13]. Because medulloblastoma is a group of heterogeneous diseases of different molecular make-up, the appropriate risk assessment and management of the disease may require the inclusion of molecular classification. Apart from a few ongoing clinical trials [14,15,16,17], risk stratification and treatment of medulloblastoma are still based on age, degree of tumor resection, and presence or absence of metastasis [18,19,20].

Medulloblastomas are commonly staged based on the Modified Chang Staging System. Despite the popularity of the modified Chang staging system across multi-centric trials in Europe and the USA, it became clear that the ‘T’ component of such a system is of limited prognostic relevance for patients with medulloblastoma [21]. Conversely, tumor histology may have a prognostic role. Severe anaplasia and large-cell variants are less frequent in adults than in children [22]. Moreover, severe anaplasia is a poor predictor of tumor recurrence and death in adults. A recent genetics-based study defined four main subgroups of medulloblastoma (i.e., WNT-activated, SHH-activated, and non-WNT/non-SHH, including Group 3 and Group 4) based on transcriptional profiling [23]. WNT-activated tumors are linked to mutations in the *CTNNB1* gene, which encodes β-catenin, and frequently have a favorable prognosis in both pediatric and adult populations [23]. Medulloblastomas of the SHH-activated subgroup are induced by mutations in genes, including *PTCH1* and *SMO*, which are integral to the Sonic Hedgehog pathway [24]. The two subgroups, Group 3 and Group 4 medulloblastomas, collectively known as non-WNT/non-SHH, lack specific subgroup-defining mutations but exhibit distinct clinical and biological features [25]. Group 3 medulloblastoma patients are more likely to present with “high-risk” characteristics, including large cell/anaplastic (LCA) histology and *MYC* amplification [22]. Group 4 medulloblastoma tumors more commonly show the presence of isochromosome 17q (i17q) [22].

We aim to review cases of adult medulloblastomas managed at a comprehensive cancer center to evaluate the use of clinical characteristics, histopathologic variables, and molecular subgrouping in predicting outcome, including overall and progression-free survival (OS and PFS, respectively).

## 2. Materials and Methods

### 2.1. Patient Population

We retrospectively reviewed the medical records of adult patients with medulloblastoma treated at King Hussein Cancer Center (KHCC) between 2007 and 2019. As a standard of care, all patients underwent complete blood count and chemistry, cerebrospinal fluid analysis, and brain and whole-spine magnetic resonance imaging (MRI) at presentation and after surgical resection.

For data collection, the following variables were retrieved from the KHCC cancer registry, patients’ files, and pathology reports: date of birth; biological sex; date of diagnosis; tumor characteristics, including tumor size, histopathologic subtype (based on review by an experienced neuropathologist, M.A.H), and molecular subtype; treatment modalities; presence of relapse; and survival status. Treatment characteristics included the extent of surgery, the interval between surgery and radiotherapy initiation, the duration of radiotherapy, and, when applicable, chemotherapy. Information on disease staging (i.e., presence of metastasis) and risk stratification was also incorporated. Molecular subgrouping was later performed on available tumor specimens (*n* = 28) by using NanoString from RNA extracted from formalin-fixed, paraffin-embedded tissue; these assessments were performed at the Hospital for SickKids, Toronto, Canada, as previously described [26]. The study protocol was approved by the KHCC Institutional Review Board (Approval # 14KHCC78). The initial phase of the approval covered the clinical and pathological findings, including the outcomes. Later on, molecular testing was conducted on cases with available paraffin blocks, and the outcome data were updated.

All patients were staged according to Chang’s staging system [27]. Accordingly, the treatment choice was based on the stratification into two risk groups: standard risk (SR) and high risk (HR). Patients with SR disease were those with less than 1.5 cm^2^ residual disease after resection and categorized as M0 by neuro-imaging and cerebrospinal fluid sampling. Patients with HR disease were those who had undergone subtotal resection with more than 1.5 cm^2^ residual tumor and/or metastatic disease.

### 2.2. Treatment

All patients underwent posterior fossa craniotomy and maximum safe resection of the tumor. The extent of tumor resection was measured by operative notes and postoperative contrast-enhanced MRI. Gross-total resection (GTR) was defined as the absence of residual tumor. Near-total resection (NTR) was defined as the presence of a residual tumor less than 1.5 cm^2^, both of which are considered for treatment purposes as complete resection, while subtotal resection (STR) was defined as the presence of residual tumor larger than 1.5 cm^2^ [28].

After surgery, all patients received an adjuvant course of CSI (36 Gy in 20 fractions), followed by a boost to the posterior fossa (18 Gy in 10 fractions). Radiation was delivered using a linear accelerator (Elekta Inc., Stockholm, Sweden). A daily dose of 1.8 Gy was administered 5 days/week via image-guided 3-dimensional (3D) conformal external-beam technique. In the case of gross metastasis in the brain or spine (M2–M3 stage), the CSI dose was escalated to 39.6 Gy, followed by a 5.4-Gy boost to gross metastatic disease. Figure 1 demonstrates the CSI plan.

Until recently, chemotherapy was not administered routinely to treat adult medulloblastoma. It was considered only in cases of metastatic disease, disease progression, or recurrence. The protocol consists of cisplatin 25 mg/m^2^ on day 1 to day 3 IV mixed with 1000 mL of 0.9% NaCl over one hour with 25 g of Mannitol and infused over one hour with Etoposide 100 mg/m^2^ day 1 to day 3 infused over one hour. The treatment is given every 4 weeks for 6 cycles. More recently, the same protocol has been administered as adjuvant chemotherapy for patients with high-risk features, including anaplastic histology, but less than in GTR cases.

### 2.3. Statistical Analysis

All data were analyzed using R (version 4.0.2). The data were presented as frequencies for categorical variables [n (%)] and median values with interquartile ranges (IQRs) for continuous variables. Overall survival (OS) is defined as the time from the start of treatment to the time of death from any cause and was censored at the date of last follow-up. Progression-free survival (PFS) is defined as the time from the start of treatment to either death, progression of disease, or development of a secondary tumor and was censored at the time of last follow-up. OS and PFS were estimated using the Kaplan–Meier method. Differences in estimated survival curves were evaluated using the log-rank test. A *p*-value ≤ 0.05 was considered statistically significant. All *p*-values were reported on 2-sided tests.

## 3. Results

### 3.1. Patient Characteristics

Fifty-three patients were included in the review. The median age of the cohort was 36.5 (30.4–42.2) years. The median follow-up was 70 (37.5–104.5) months. Males outnumbered females 29 (54.7%) to 24 (45.3%). GTR or NTR, considered as complete resection, was performed in 43 (81%) patients; STR in 9 (17%) patients; and biopsy in 1 (2%) patient (Table 1).

Tumors were located laterally in 40 patients (75.5%) and centrally in 13 (24.5%). The median tumor size for the cohort was 4.0 (3.0–4.8) cm^2^. In terms of histopathology, tumors were equally distributed between classic (45.3%) and desmoplastic/nodular (D/N; 45.3%), with only five (9.4%) cases of large-cell/anaplastic (LC/A) medulloblastoma. The tumor specimens included three molecular subtypes: SHH-activated (78.6%), WNT-activated (10.7%), and Group 4 (10.7%). No Group 3 tumors were identified. Metastasis was present in 22.6% of cases, and 41.4% of cases were categorized as HR.

The median interval from surgery to the start of radiotherapy was 6.1 (4.8–9.5) weeks, and the median duration of radiation treatment was 44.6 (41.6–47.1) days.

### 3.2. Survival

Among the patients included in the review, only 20 (37.7%) died. Table 2 shows the survival differences among the study’s participants. The mean 3-year, 5-year, and 10-year OS of participants were 85% (75–95%), 74% (62–87%), and 50% (33–75%), respectively. Overall survival significantly differed based on the extent of surgery (*p* = 0.017), M stage (*p* = 0.009), and risk status (*p* = 0.001). Relapses were recorded in 15 (28.3%) cases. The 3-year, 5-year, and 10-year PFS were 81% (71–92%), 75% (63–88%), and 66% (5–83%), respectively. Progression-free survival also significantly differed based on the extent of surgery (*p* = 0.008) and risk status (*p* = 0.012).

In terms of histology, 5-year survival for the classic, D/N, and LC/A medulloblastomas were 76% (60–97%), 74% (58–94%), and 53% (21–100%) for OS and 87% (74–100%), 65% (48–88%), and 60% (29–100%) for PFS, respectively, none of which correlated with OS or PFS. Three patients with WNT-activated medulloblastoma are alive and free of disease at 5 years; median survival was not reached for three patients with Group 4 disease. The mean 5-year OS and PFS were 65% (47–90%) and 62% (44–87%), respectively, in 22 patients with SHH-activated medulloblastoma. However, molecular subgrouping did not correlate with OS or PFS. Figure 2 and Figure 3 demonstrate the Kaplan–Meier estimates for the included groups.

## 4. Discussion

Our study summarizes the findings in a cohort of adult patients diagnosed with medulloblastoma, a rarely occurring tumor in this age group. Among our cohort of adult patients with medulloblastoma, OS ranged from 85% at 3 years to 50% at the 10-year mark. Also, PFS ranged from 81% at 3 years to 66% at the 10-year mark. Overall survival was significantly associated with the extent of surgery, risk status, and M stage. The earlier two were also significantly associated with PFS. It appears that neither histological subtype nor molecular subgrouping affects survival. However, such might be the case due to the small number of cases. Time from surgery to radiotherapy and duration of radiotherapy did not impact survival, either.

Because medulloblastoma rarely arises in adults, prospective data in this population is scarce, and oncologists tend to follow treatment guidelines for pediatric medulloblastoma to treat adults with this disease. In the last few decades, survival has increased, reaching approximately 70% [29,30]. Our results are in line with those estimates, as we demonstrated a 5-year mean OS and PFS of 74% and 75%, respectively.

Survival outcomes in medulloblastoma differ between children and adults, though initial overall survival rates are often similar, typically ranging between 75 and 85% for both groups [29,31]. However, long-term survival differs, with adults generally experiencing worse outcomes after about four years post-diagnosis [29]. Children, particularly those with favorable molecular subtypes, such as WNT-activated tumors, tend to have better survival prospects [10,32]. In contrast, LC/A histology and extensive metastasis are more detrimental in adults, though GTR improves prognosis [31,32]. Molecular subtypes also play a role in survival differences; while the WNT subtype is favorable in both groups, Group 3 tumors tend to have worse outcomes, especially in children [10]. Additionally, adults are more likely to develop SHH-activated tumors, which can vary in prognosis based on further molecular subtypes [32]. Despite similar treatment strategies, like radiotherapy, the extent of disease at diagnosis, such as brainstem involvement and distant metastasis, remains a critical factor in determining survival outcomes, with children showing better long-term survival when these factors are favorable [29,30].

The long-term implications of medulloblastoma treatments vary by age group due to developmental factors and the unique nature of disease progression and treatment side effects in these populations [33,34,35]. Palmer et al. assessed the processing speed (PS), working memory (WM), and broad attention (BA) in patients aged 3–21 years old [33]. Pediatric patients were particularly vulnerable to treatment-related neurocognitive decline due to ongoing brain development [33]. Younger age and higher treatment intensity (e.g., high-risk classification) were significant predictors of worse outcomes [33]. Furthermore, Papini et al. reported that long-term survivors of childhood medulloblastoma continue to face significant risks of neurocognitive impairment, particularly in memory and task efficiency, with these issues linked to chronic health conditions and reduced functional independence [34]. Unlike pediatric patients, adult patients demonstrated some initial improvements in certain cognitive functions, such as processing speed and verbal fluency, during early follow-up after treatment [35]. However, other domains, such as working memory, were impaired post-treatment and remained stable or declined slightly over time [35]. Treatment significantly impacts the health-related quality of life (HRQoL) in both pediatric and adult medulloblastoma patients, though the effects vary by age. Pediatric patients experienced notable declines in HRQoL, particularly in social and role functioning, with younger children showing greater vulnerability due to ongoing cognitive development. In contrast, adult patients initially showed improvements in HRQoL but faced declines in cognitive functioning after 30 months [33,35].

Surgical resection and adjuvant radiotherapy are considered the mainstay treatments for medulloblastoma in both the adult and pediatric populations [12,19]. Recently, advances in neuroimaging and neurosurgery have improved tumor resection and decreased surgical morbidity and mortality [30,36]. This is well documented in this cohort, in whom the extent of surgical resection in the form of complete resection was associated with better OS and PFS.

Similarly, radiation oncology has advanced dramatically in the past few decades. New machines and techniques enable radiation oncologists to deliver radiation more precisely to the target volume, thereby sparing healthy tissues [37]. For example, irradiation of the ovaries with the lower spinal beam can cause premature ovarian failure and subsequent sterility [38]. These drastic sequelae can be avoided using novel radiotherapy techniques and ovarian transposition [39,40].

Early initiation of radiotherapy within 4–6 weeks after surgery may confer better disease control [41]. Moreover, Abacioglu et al. demonstrated that adult patients with medulloblastoma who start radiotherapy 3–6 weeks after surgery have the highest 5-year disease-free survival [42]. In our cohort, the median interval between surgery and radiotherapy was 6 weeks; however, differences in the surgery-to-radiation interval did not translate into significant survival differences among our cohort. Del Charco et al., in their 30-year review, showed the detrimental effects on local control when the duration of radiotherapy was more than 45 days [43]. However, the median duration of radiotherapy was 44.6 days in our cohort, which did not affect survival.

The role of chemotherapy in adult medulloblastoma is debatable. In pediatric patients with medulloblastoma, chemotherapy serves to reduce the dose of radiotherapy administered, thereby alleviating the effect of radiation on skeletal growth and reducing the risk of secondary malignancy. The potential effects of radiotherapy are less concerning in adults. Brandes et al., in their prospective study of 36 patients treated on SR or HR regimens, including two courses of chemotherapy (i.e., cisplatin–etoposide–cyclophosphamide) before and after CSI [44], found no statistical difference in terms of disease control or survival. Of note, the long-term results revealed an increased recurrence rate in patients with SR medulloblastoma, implying a potential role of chemotherapy [44,45].

Friedrich et al. studied patients who received eight cycles of chemotherapy after radiotherapy according to the HIT 2000 protocol (lomustine, vincristine, cisplatin) and demonstrated that grade 3 hematotoxicity occurred in 55% of patients and grade 2–3 neurotoxicity occurred in about 74% [46]. Only half of the patients enrolled completed the recommended regimen. The German Neuro-oncology Working Group study (NOA-7) prospectively investigated the impact of radiotherapy plus vincristine on adult patients with medulloblastoma [47]. About 65% of patients withdrew from the study due to maintenance chemotherapy-induced toxicity, including polyneuropathy and leukopenia. Silvani et al. studied the impact of chemotherapy followed by radiotherapy on 28 patients with medulloblastoma, irrespective of risk category [48]. In their cohort, the chemotherapeutic regimen was well tolerated; however, it did not reduce recurrence, particularly distant recurrence. Moots et al. studied the value of neoadjuvant chemotherapy (cisplatin, etoposide, cyclophosphamide, and vincristine) followed by radiotherapy [49]. The authors found that neoadjuvant chemotherapy did not affect the patient’s ability to complete radiotherapy; however, the survival outcomes were dismal, leading to the early closure of the study.

In contrast, Kann et al. retrospectively evaluated the role of adjuvant chemotherapy by using the National Cancer Database Registry [50]. They found that combined radiotherapy and chemotherapy results in better 5-year OS than radiation alone, but only in specific groups of patients: those with M0 disease (no detectable metastasis), those who received higher-than-standard doses of radiation, or those with both of these characteristics (with M0 disease and received higher-than-standard doses of radiation). Similarly, a meta-analysis by Kocakaya et al. showed that patients who received chemotherapy and radiotherapy had significantly better prognosis than those who received only radiotherapy or only chemotherapy at recurrence [51].

To date, no conclusive evidence is available on whether adjuvant chemotherapy shows any benefit in adults with medulloblastoma, specifically those with SR disease. For patients with HR medulloblastoma, upfront chemotherapy appeared to be tolerated, but the evidence must be considered with caution. There is a paucity of prospective studies on various aspects of chemotherapy, including timing of chemotherapy, type of chemotherapy, and the role of chemotherapy in the four molecular subtypes.

In our study, 11 (26%) patients experienced disease relapse after a median of 27 months. Unlike pediatric medulloblastoma, which is associated with recurrence within the first 24 months, adults with medulloblastoma have late recurrences [52,53]. Thus, this subset of patients requires meticulous long-term follow-up. Salvage options for recurrent disease include second surgery and chemotherapy, in addition to focal re-irradiation [41,54]. Also, high-dose chemotherapy and autologous stem cell transplantation have shown promising results [55].

Per the latest WHO 2021 CNS Tumor Classification, medulloblastoma is defined by either histology or molecular constitution [56]. Its histologic subtypes include the classic, D/N, LC/A, and medulloblastoma with extensive nodularity variants. Classic medulloblastoma is the most common histologic subtype in literature, and LC/A and medulloblastoma with extensive nodular variants are the rarest [6,24]. Among our cohort, classic and D/N variants were the most common. The prognostic value of histologic variants is inconsistent in adult medulloblastoma literature. Zhang et al. and de Haas et al., among others, have reported an association among histological subtypes, particularly D/N medulloblastoma, and survival outcomes [12,57,58,59,60,61,62]. In contrast, Chan et al., Friedrich et al., and others have observed no such associations [46,63,64,65,66]. Our findings are in line with the latter group, as we found no significant differences in OS or PFS among the histological subtypes. These intra- and inter-differences in the prognostic value of histology are often attributed to methodologic heterogeneity mediated by sample size.

Transcriptional-array profiling and, more recently, methylation profiling assays now enable investigators to refine treatment stratification into four distinct molecular subgroups: WNT-activated, SHH-activated, non-WNT/non-SHH (Group 3 and Group 4). Of the aforementioned subtypes, SHH-activated is the most common, accounting for 60% of adult cases, and Group 3 adult medulloblastoma is extremely rare. Several prognostic differences have been observed between adult and pediatric patients with medulloblastoma [6]. WNT-activated and Group 4 tumors are associated with worse survival outcomes in adults than in pediatric patients [6,7]. This difference stems from heterogeneity in molecular characteristics across the two age groups [17,18,45]. For example, half of adult WNT-activated tumors lack the *CTNNB1*-activating mutations [67]. Compared to the pediatric disease, adult medulloblastoma with nuclear ß-catenin and chromosome 6q deletion is not associated with a better prognosis [6]. In addition, a chromosomal loss of 10q is prognostic in adult SHH-activated tumors but not in pediatric tumors [23]. Therefore, extrapolation from pediatric series may have severe limitations in adults. Amongst our cohort, we could not detect a difference in survival between molecular groups due to the brief follow-up period and the relatively small cohort size. Ongoing trials (NCT01878617 and NCT01708174) are aimed at testing the efficacy of novel treatments based on molecular subgroups and risk stratification [68,69]. These studies may provide personalized treatment to improve survival and decrease toxicity.

Three limitations might have affected the generalizability of our findings: First, the retrospective nature of our design includes an associated potential selection bias. Second, the number of patients was relatively small, which might have affected the rigor of our analysis. Third, adult medulloblastoma tends to recur late after treatment, which entails longer follow-up than included in this report.

## 5. Conclusions

Treatment of adult medulloblastomas is influenced by pediatric experience. Although surgical resection and radiotherapy are crucial for the clinical management of this disease, the use of chemotherapy lacks evidence-based guidelines. Some reports have shown the benefit of chemotherapy, specifically in patients with HR disease. Our results revealed poor survival in the HR group, who may benefit from intensification with chemotherapy. In the case of recurrence, the prognosis of adult medulloblastoma is dismal; salvage options should be further investigated. The new molecular subgrouping signals a new era in medulloblastoma management. Although we did not detect any impact of the subgroups on disease prognosis in our cohort, larger studies of adults with medulloblastoma may better stratify the cohort of patients.

## Figures and Tables

**Figure 1 cancers-16-03609-f001:**
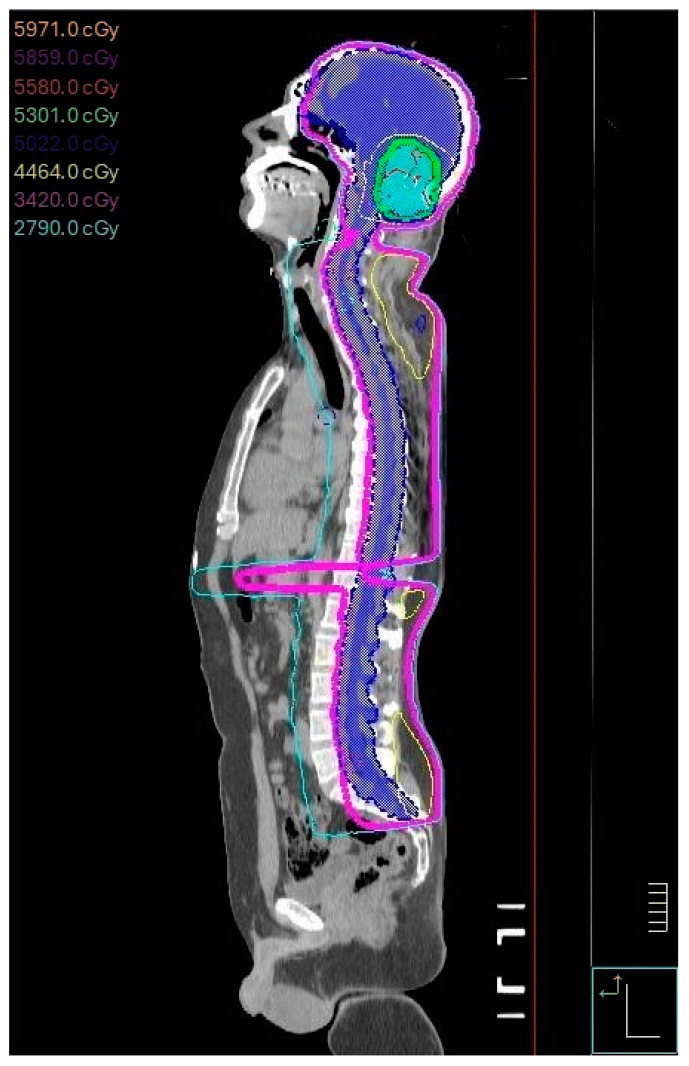
Craniospinal irradiation (CSI) plan as used in radiation therapy planning.

**Figure 2 cancers-16-03609-f002:**
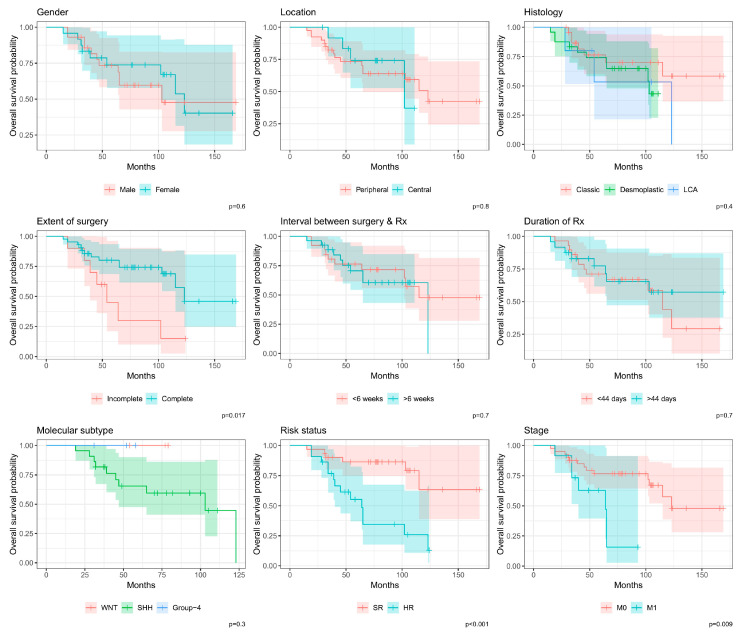
Kaplan–Meier graph of overall survival among different groups. LCA: Large-cell/anaplastic variant; Rx: radiotherapy; SR: standard risk; HR: high risk; M0: no metastasis; M1: metastasis present; SHH: Sonic Hedgehog; WNT: Wingless Integration 1.

**Figure 3 cancers-16-03609-f003:**
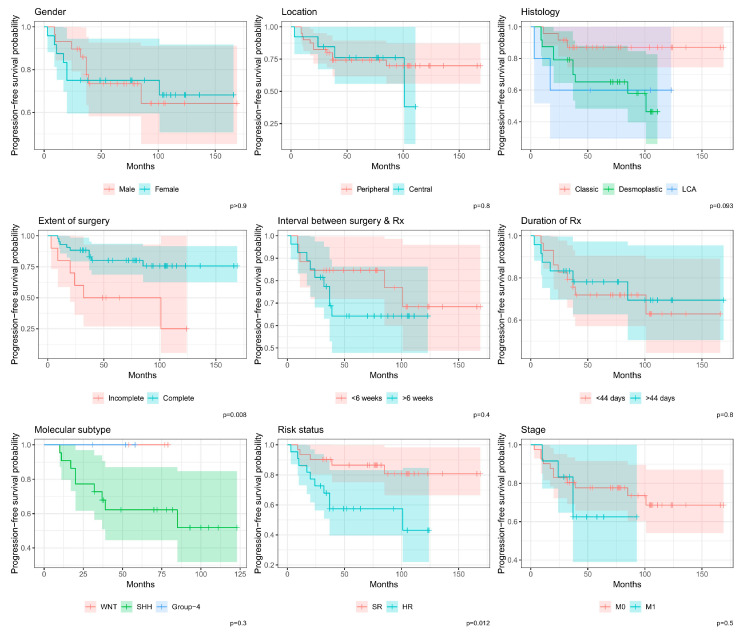
Kaplan–Meier graph of progression-free survival among different groups. LCA: large-cell/anaplastic variant; Rx: radiotherapy; SR: standard risk; HR: high risk; M0: no metastasis; M1: metastasis present; SHH: Sonic Hedgehog; WNT: Wingless Integration 1.

**Table 1 cancers-16-03609-t001:** Patient demographics and tumor and treatment characteristics.

Variable	Mean (%)
**Age at diagnosis** All patients, median age (IQR)	36.5 (30.4–42.2) y
<25 y	2 (3.8)
≥25 y	51 (96.2)
**Sex**	
Male	29 (54.7)
Female	24 (45.3)
**Location of tumor**	
Peripheral	40 (75.5)
Central	13 (24.5)
**Median tumor size (IQR)**	4.0 (3.0–4.8) cm^2^
**Histopathology**	
Desmoplastic/Nuclear	24 (45.3)
Classic	24 (45.3)
Large-cell/anaplastic	5 (9.4)
**Extent of surgery**	
Less than gross-total resection	10 (18.9)
Gross-total resection	43 (81.1)
**Risk stratification**	
Standard risk	31 (58.5)
High risk	22 (41.5)
**Disease stage at diagnosis**	
M0	41 (77.4)
M1	12 (22.6)
**Molecular subgroup**	
WNT-activated	3 (10.7)
SHH-activated	22 (78.6)
Group 4	3 (10.7)
**Interval between surgery and ** **start of radiation therapy**	
Median interval (IQR)	6.1 (4.8–9.5) wk
≤6 wk	26 (49.1)
>6 wk	27 (50.9)
**Duration of RT**	
Median duration (IQR)	44.6 (41.6–47.1) days
≥44 days	24 (45.3)
<44 days	29 (54.7)
**Median follow-up (IQR)**	70.0 (37.5–104.5) months
**Mean survival (95% CI)**	
Progression-free	125.0 (106.3–143.7) months
Overall	119.4 (93.5–130.4) months
**Relapse**	
No	38 (71.7)
Yes	15 (28.3)
**Status**	
Alive	33 (62.3)
Dead	20 (37.7)

**Table 2 cancers-16-03609-t002:** Five-year OS and PFS of adult patients with medulloblastoma.

Demographic Trait	Overall Survival Mean (%)	*p*-Value *	Progression-Free Survival Mean (%)	*p*-Value *
**Age at diagnosis**		NA		NA
<25 y	-		-	
≥25 y	72 (61, 87)		73 (62, 87)	
**Sex**		0.625		0.982
Male	73 (58, 93)		73 (58, 93)	
Female	74 (58, 94)		75 (60, 94)	
**Tumor location**		0.817		0.788
Peripheral	73 (61, 89)		74 (62, 89)	
Central	74 (53, 100)		76 (56, 100)	
**Histology**		0.418		0.093
Classic	76 (60, 97)		87 (74, 100)	
Desmoplastic/Nuclear	74 (58, 94)		65 (48, 88)	
Large cell/Anaplastic	53 (21, 100)		60 (29, 100)	
**Extent of resection**		**0.017**		**0.008**
Incomplete	45 (21, 96)		50 (27, 93)	
Complete	80 (69, 94)		80 (69, 94)	
**Risk status**		**0.001**		**0.012**
Standard risk	86 (75, 100)		87 (75, 100)	
High risk	55 (37, 83)		57 (40, 83)	
**Disease stage at diagnosis**		**0.009**		0.475
M0	77 (64, 91)		78 (66, 92)	
M1	63 (39, 100)		62 (39, 100)	
**Molecular subtype**		NA		NA
WNT-activated	100 (100, 100)		100 (100, 100)	
SHH-activated	65 (47, 90)		62 (44, 87)	
Group 4	-		-	
**Interval between surgery and radiation therapy**		0.679		0.373
≤6 wk	76 (61, 95)		85 (72, 100)	
>6 wk	71 (54, 92)		64 (48, 86)	
**Radiation therapy duration**		0.698		0.778
<44 days	71 (56, 90)		72 (57, 91)	
≥44 days	77 (62, 97)		78 (63, 97)	
**Total sample**	74 (62, 87)	NA	75 (63, 88)	NA

* Significant *p*-values are formatted in bold. Abbreviations: NA—not applicable.

## Data Availability

All data/datasets associated with this project will be provided at a reasonable request from the corresponding author.

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
