# Peer review of "Clinical and Molecular Characteristics and Outcome of Adult Medulloblastoma at a Tertiary Cancer Center"

_cancers, 2024, doi:10.3390/cancers16213609_

Round 1
Reviewer 1 Report
Comments and Suggestions for Authors
1. In line 72, the authors write "WNT-activated, SHH-activated, Group 3, and Group 4", please explain what Group 3 and Group 4 mean.
2. The authors provide information on radiotherapy in days, but do not indicate the radiation dose. Please add it.
3. I wonder if the survival data differs from those for children? Maybe add this to the Discussion section.
4. Formulate, is the treatment strategy for medulloblastoma in adults different? In what way exactly?
Author Response
1. In line 72, the authors write "WNT-activated, SHH-activated, Group 3, and Group 4," please explain what Group 3 and Group 4 mean.
Response: We thank you for your comment, a sentence regarding the explanation of group 3 and 4 were added as required. See page 2, line 77-90.
2. Add the radiation dose to the section discussing radiotherapy.
Response: We thank you for your comment, the radiation dose is written in the treatment section. See page 3, lines 130-133.
3. Consider comparing the survival data to those for children and add this to the Discussion section.
Response: We thank you for your comment, a paragraph comparing the survival outcomes between adults and pediatric was added as required. See page 9-10, lines 213-226.
4. Clarify how the treatment strategy for medulloblastoma in adults differs from that in children. In what way exactly?
Response: We thank you for your comment, a paragraph comparing the survival outcomes between adults and pediatric was added as required. See page 9-10, lines 213-226.
Reviewer 2 Report
Comments and Suggestions for Authors
This is a retrospective review of single-center case series of medulloblastoma.
Although surgical extent was defined as GTR, STR and NTR, table 1 did not show STR and NTR. And residual tumor larger than 1.5 cm2 is not acceptable definition of NTR, those should be partial resection.
62.3% of the patients are still alive, so OS is not good indicator to evaluate risk factors. The treatment strategy of radiation seem to be consistent, however, the treatment strategy of chemotherapy is inconsistent. So these treatment strategies could not be evaluated as the outcome of these patients series. If all patients received same radiation therapy, radiation therapy could not be the factor to determine the outcome.
Risk factors should be evaluated using single viariate and multi variate analysis.
Author Response
1. Although the extent of surgery was defined as GTR, STR, and NTR, Table 1 did not show STR and NTR. Residual tumors larger than 1.5 cm² are not acceptable definitions of NTR; this should be defined as partial resection.
Response: Thank you for your comment. Please note that for treatment consideration, GTR and NTR are grouped together under complete surgical resection, whereas STR and biopsy are under incomplete surgical resection.
2. 62.3% of patients are still alive, so overall survival (OS) is not a good indicator to evaluate risk factors.
Response: Dear respected reviewer, this is rather an adherence to cancer research conventions. PFS and OS are considered classical endpoints of survival (DOI: 10.3978/j.issn.2218-6751.2011.12.08). Indeed, OS doesn’t differentiate for causes of death (that’s where PFS acts adding granularity and focus) and in our case, OS would be more meaningful at a later follow-up date. However, a more focused approached, such as using patient reported outcomes, is unfortunately hindered by the limitations of data and data collection practices at KHCC.
Data regarding adult medulloblastoma is rare to begin with. Thus, a full exploration of any survival-related outcome variable might be necessary to expand the fullest of our data and showcase the clinical and mortality characteristics of the included sample. Please note that this manuscript isn’t a demonstration of the superiority or (non-inferiority) of any treatment regimen, but rather a descriptive showcasing of the KHCC experience with adult patients with medulloblastoma.
3. The treatment strategy of radiation seems consistent, but the strategy of chemotherapy is inconsistent. This makes it difficult to evaluate the outcomes.
Response: We thank you for your comment, the chemotherapy strategy was further explained in the treatment section. See page 4, lines 143-147.
4. If all patients received the same radiation therapy, radiation alone cannot be the factor determining the outcome.
Response: Thank you for your comment. True, radiotherapy was not among the factors associated with outcome in this cohort of patient.
Reviewer 3 Report
Comments and Suggestions for Authors
Clinical and molecular characteristics and outcome of adult medulloblastoma at a tertiary cancer center
The manuscript is very interesting and well-written. However, after reading the whole manuscript, the introduction needs to be improved, as a general recommendation, and more details need to be added to the statistical analyses since it is a retrospective study.
Authors can find my suggestions, recommendations, and concerns as follows:
Introduction: The section is well-written but it needs to be improved. Indeed, the authors need to add more details about Medulloblastoma in adults in terms of clinical characteristics and epidemiology. Indeed, the authors underlined that is a rare disease in the adult population, but I advise adding the prevalence. Please, add more info about the genetic-based classification as described by Remke (2011). Moreover, the authors need to add a brief paragraph about previous studies. The hypotheses need to be stated in a more specific way.
Methods: The patient population is described in a detailed way. About the statistical analyses, the authors need to indicate if they applied a correction, and if a Dunn test was calculated for the Kruskal-Wallis test. Moreover, in the tables, I advise reporting the value of the test. Moreover, it is not clear when you applied MW or KW test. Similarly, the same for Chi-sqr. Please, revise the tables and the text.
Results: the section is interesting, but please see the commentaries for statistical analyses. A brief caption in fig.2 is needed.
The discussion is well-written and informative. The authors are able to integrate their findings with previous published evidence. However, I advise to highlight more of the differences between Medulloblastoma in children and adults.
Author Response
1. The introduction needs to be improved by adding more details about medulloblastoma in adults in terms of clinical characteristics and epidemiology. Please also include the prevalence.
Response: we thank you for your comment, the required changes in the introduction were made. See page 2, lines 47-50, lines 53-57, lines 77-90.
2. Add more information about the genetic-based classification as described by Remke (2011).
Response: we thank you for your comment, details regarding the genetic based classification was added in the introduction section. See page 2, lines 77-90.
3. The authors need to add a brief paragraph about previous studies. The hypotheses need to be stated in a more specific way.
Response: we thank you for your comment, the required paragraph was added in the discussion section. See page 10, lines 239-272.
4. Indicate whether a correction was applied in the statistical analysis, and if a Dunn test was calculated for the Kruskal-Wallis test.
Response: Dear respected reviewer, upon revision of the manuscript, we realized that the conducted statistics only pertained to survival analysis. No chi-squared or (MWU or KW) were applied as those did not have a relevant outcome variable. We apologize for including this artifact that may have cause a misunderstanding. See page 10, lines 151-162.
5. In the tables, report the value of the tests and specify when IMW or KW tests were applied. The same applies to Chi-sqr. Please revise the tables and the text.
Response: Dear respected reviewer, upon revision of the manuscript, we realized that the conducted statistics only pertained to survival analysis. No chi-squared or (MWU or KW) were applied as those did not have a relevant outcome variable. We apologize for including this artifact that may have cause a misunderstanding. The text of the statistical analysis section is now edited extensively to showcase the details of what was conducted. See page 10, lines 151-162.
6. The results section is interesting, but more details on statistical analyses are required. A brief caption in figure 2 is needed.
Response: The text of the statistical analysis section is now edited extensively to showcase the details of what was conducted. The figures are edited per the reviewer’s request. See page 10, lines 151-162. See figure 2.
7. The discussion is well-written, but highlight more differences between medulloblastoma in children and adults.
Response: we thank you for your comment, a paragraph comparing the differences between medulloblastoma in children and adults. See page 10, lines 151-162.
Reviewer 4 Report
Comments and Suggestions for Authors
In this manuscript titled “Clinical and molecular characteristics and outcome of adult medulloblastoma at a tertiary cancer center”, Almousa and his colleagues reported the clinical characteristics, prognostic factors, and treatment outcome of a cohort of adult patients with medulloblastoma. Here are my comments and concerns related to this manuscript.
-The main concern of this manuscript is that the authors intend to describe the clinical and molecular characteristics of medulloblastoma in adults. However, It is unclear on what basis the authors classified the medulloblastoma groups studied.
-There is no information in the introduction section related to the molecular characteristics related to medulloblastoma in adults. In fact, the results section does not contain information related to the molecular characteristics related to medulloblastoma in adults.
-What criteria were considered to classify the four main subgroups of medulloblastoma (WNT-activated, SHH-activated, Group 3, and Group 4). In this regard, the authors do not show the histological and Immunohistochemical analyses.
-The materials and methods need to be improved.
-In the materials and methods section the authors describe the total RNA extraction from formalin-fixed, paraffin-embedded tissue. However, they say that a biopsy is obtained from 1 patient. It is unclear what total RNA is used for in this study.
-In addition to radiotherapy treatment, what other treatment did patients with medulloblastoma receive?
-This manuscript has results focused primarily on overall survival.
-There are some typos and words with grammatical errors.
Comments on the Quality of English LanguageMinor editing of English language required.
Author Response
1. It is unclear on what basis the authors classified the medulloblastoma groups. Provide clarification on how the molecular characteristics were used to classify the groups studied.
Response: Thank you for your comment. The methods by which the medulloblastoma subtypes were classified are mentioned in the manuscript file lines 109-112.
2. There is no information in the introduction related to the molecular characteristics of medulloblastoma in adults. This needs to be addressed.
Response: we thank you for your comment, a paragraph regarding the molecular characteristics in the introduction section was added. See page 2, lines 77-90.
3. Clarify the criteria used to classify the four main subgroups (WNT-activated, SHH-activated, Group 3, and Group 4).
Response: we thank you for your comment, a paragraph regarding the molecular characteristics in the introduction section was added. See page 2, lines 77-90.
4. Improve the materials and methods section, particularly in relation to the RNA extraction process and the description of the biopsy.
Response: Thank you for your comment. This has now been updated and a reference to the methods used for RNA extraction and Nonostring testing is added to the methods section (Page-3, line 112).
5. It is unclear what the total RNA is used for in this study. Please clarify.
Response: Thank you for your comment. Please note that the paraffin shaves of samples were sent to Hospital for Sick Kids for the Nanostring analysis as explained in the methods section. A reference has been added to explain the test method that was used.
6. In addition to radiotherapy, what other treatments did patients receive? Please specify.
Response: we thank you for your comment, the other treatments that the patient was further explained in the treatment section, see page 3-4, lines 124-147.
7. The manuscript focuses mainly on overall survival; expand the discussion on other outcomes if possible.
Response: We thank you for your comment, a paragraph discussing other outcomes of medulloblastoma was added. See page 10, lines 239-272.
Round 2
Reviewer 2 Report
Comments and Suggestions for Authors
I found significant improvements in this manuscript.
So this is acceptable now.
Reviewer 3 Report
Comments and Suggestions for Authors
The manuscript was improved, and I agree with the statistical analysis.
Reviewer 4 Report
Comments and Suggestions for Authors
I have no comments, my suggestions were addressed.
Comments on the Quality of English LanguageThere are some typos in the manuscript.